# Rainfall Investigation by Means of Marine In Situ Gamma-ray Spectrometry in Ligurian Sea, Mediterranean Sea, Italy

**Dionisis L. Patiris** [1,*] **, Sara Pensieri** [2] **, Christos Tsabaris** [1] **, Roberto Bozzano** [2] **, Effrossyni G. Androulakaki** [1] **, Marios N. Anagnostou** [3,4] **and Stylianos Alexakis** [1]

[1]  Hellenic Centre for Marine Research, Institute of Oceanography, 19013 Anavyssos, Greece; tsabaris@hcmr.gr (C.T.); frosso.androulakaki@hcmr.gr (E.G.A.); salexakis@hcmr.gr (S.A.)

[2]  Institute of Anthropic Impacts and Sustainability in Marine Environment, National Research Council of Italy, 16149 Genova, Italy; sara.pensieri@cnr.it (S.P.); roberto.bozzano@cnr.it (R.B.)

[3]  Institute of Environmental Research and Sustainable Development, National Observatory of Athens, 15236 Old Penteli, Greece; sifneos@live.com

[4]  Department of Naval Architecture, School of Engineering, University of West Attica, 12243 Athens, Greece

*  Correspondence: dpatiris@hcmr.gr; Tel.: +30-22-9107-6408

**Abstract:** Marine in situ gamma-ray spectrometry was utilized for a rainfall study at the W1M3A observing system in Ligurian Sea, Mediterranean Sea, Italy. From 7 June to 10 October 2016, under-water total gamma-ray counting rate (TCR) and the activity concentration of radon daughters $^{214}$Pb, $^{214}$Bi and potassium $^{40}$K were continuously monitored along with ambient noise and meteorological parameters. TCR was proven as a good rainfall indicator as radon daughters' fallout resulted in increased levels of marine radioactivity during and 2–3 h after the rainfall events. Cloud origin significantly affects TCR and radon progenies variations, as aerial mass trajectories, which extend upon terrestrial areas, result in higher increments. TCR and radon progenies concentrations revealed an increasing non-linear trend with rainfall height and intensity. $^{40}$K was proven to be an additional radio-tracer as its dilution was associated with rainfall height. $^{40}$K variations combined with $^{214}$Bi measurements can be used to investigate the mixing of rain- and seawater. In comparison with measurements in the atmosphere, the application of marine in situ gamma-ray spectrometry for precipitation investigation provided important advantages: allows quantitative measurement of the radionuclides; $^{40}$K can be used, along with radon daughters, as a radio-tracer; the mixing of rain- and seawater can be associated with meteorological parameters.

**Keywords:** marine radioactivity; in situ gamma-ray spectrometry; precipitation; radon; potassium $^{40}$K; radio-tracers

## 1. Introduction

Gamma-ray spectrometry is a well-established technique for wide range of radio-tracing applications in the environment. Regarding precipitation studies, the most utilized radionuclides are the naturally occurring radon $^{222,220}$Rn and radon progenies ($^{214}$Pb, $^{214}$Bi) [1–9], as well as the cosmogenic $^{7}$Be [1,10–16]. Radon progenies are easier to detect than $^{7}$Be, either by sampling and measurement of rainwater or snow quantities or by in situ gamma-ray spectrometers exposed to precipitation, as they emit several gamma rays of high emission probability in a wide energy range (350–2200 keV). Radon as inert gas emanates from terrestrial areas and in the atmosphere decays producing radon progenies. Radon progenies attach onto atmospheric aerosols and, consequently, take part in cloud formation processes. Thus, in precipitation water, they may be found in elevated levels, making their detection feasible with proper uncertainty for further statistical analysis. Involved in cloud formation processes and precipitation, radon progenies were utilized as tracers in several atmospheric studies. Both $^{214}$Pb and $^{214}$Bi activity concentration and the ratio of $^{214}$Pb/$^{214}$Bi have been proposed to estimate the age of precipitation, the time

span of in-cloud droplet movement processes, falling time and cloud height as well as the precipitation rate [2,5–8,17–20].

Although most precipitation events occur upon the oceans, few works have been published investigating them in the marine environment. For such surveys, the marine in situ gamma-ray spectrometry is the optimum approach, as the need for continuous monitoring is critical to collect data before, during and after a precipitation event, and the operational cost of alternative sampling missions is high. During the last few decades, considerable progress has been made in the field of technological development of in situ underwater gamma-ray spectrometers [21–25], resulting in several applications of in situ radio-tracing techniques in the marine environment [26–30]. From the few published works of seawater monitoring during precipitation events, radon progenies ($^{214}$Pb, $^{214}$Bi) have been indicated as promising tracers since precipitation events result in a significant increase in their activity concentration [31,32]. Moreover, the open sea is an ideal environment to utilize in situ gamma-ray spectrometry for precipitation investigation. In comparison with the atmosphere, it provides an almost steady environment (e.g., low daily temperature variations, no moisture, atmospheric pressure, interferences), much lower radiation background level and the possibility of quantitative estimation of activity concentration [33–36]. Additionally, the background level depends mostly on the concentration of the radioactive isotope of $^{40}$K, which is rather constant in the absence of radiological incidences and/or interferences with other water masses (e.g., submarine groundwater and river discharge), so fluctuations of the total detected gamma rays can be directly associated with precipitation events. Furthermore, $^{40}$K itself can be utilized as a new potential radio-tracer of precipitation due to its high activity concentration in seawater, which allows it to be quantitatively estimated with high precision. Coupling of radon progenies and $^{40}$K has already been used to study the mixing processes of water masses of different origins [37–41] as well as the interferences of sediment re-suspension with the seawater radon progenies measurements [27,30].

The aim of the present work is to realize a prolonged (more than 5 months) and continuous in situ monitoring of gamma-ray-emitting radionuclides activity concentration in the marine environment. Simultaneously, the radiological condition of the marine environment is surveyed, and natural radionuclides are used as tracers in the investigation of precipitation over the seas. Many natural radionuclides are involved in the water cycle, so they can be exploited in radio-tracing approaches to reveal potential associations with precipitation parameters and new methods to investigate the water cycle, which is directly related with the climate variability. In this work, radon progenies and $^{40}$K associations with precipitation parameters are presented. Additionally, the dependency of their activity concentration in seawater after rainfall with the cloud origin and aerial masses trajectories is revealed. The fall out of natural radionuclides possess no threat for the environment; however, it can be used to further understand the complicated transport of harmful anthropogenic radionuclides emitting into the atmosphere during accidental releases. Therefore, in this work, new technological advances are used (sensors, observing systems and monitoring stations) to simultaneously survey the radiological state of the marine environment and highlight the possibility of using natural radionuclides in water cycle studies. The work highlights the capability of the marine monitoring network in the Mediterranean Sea to provide multi-use data for marine surveillance and investigation of ocean–atmosphere interaction.

## 2. Study Area, Materials and Methods

The experiment was realized in the Ligurian Sea, which is a semi-closed basin in the northwest Mediterranean Sea from 7 June 2016 to 10 October 2016. The area is surrounded by the Liguria region from the north and Corsica Island from the south. The W1M3A observing system [42,43] is permanently moored 80 Km from the Ligurian coast and on the route between Genoa and Cape Corse. For the experiment, the data collected by the compact weather station installed on the upper mast of the W1M3A observatory

and several bio-chemical sensors deployed at the sea surface and beyond were used. The area is also covered by one weather radar located on mount Settepani (Figure 1). Technological, operational and technical information about the W1M3A observing system and the implementation of the experimental part of this work is described in details elsewhere [31].

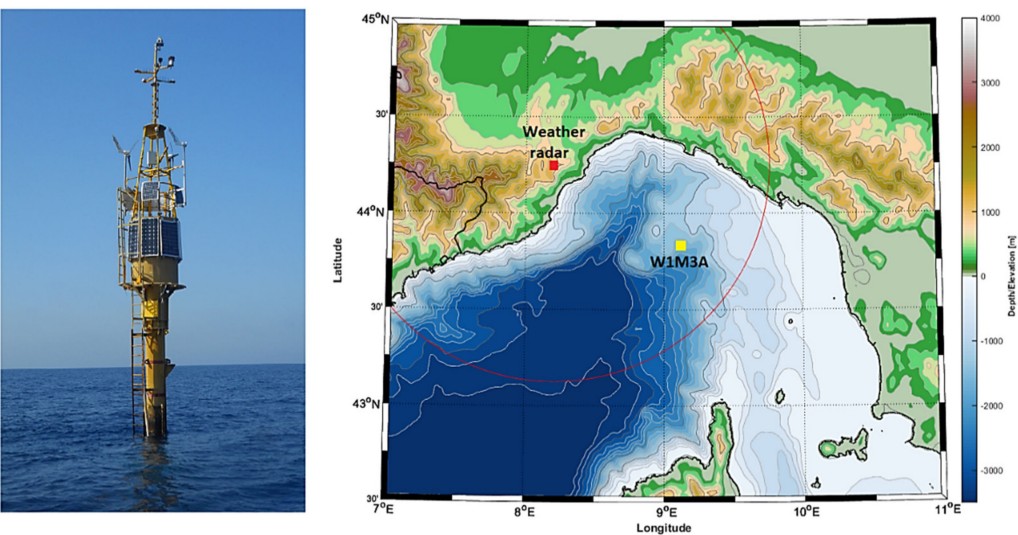

**Figure 1.** A photo of the surface buoy of the W1M3A observing system and the map of the Ligurian basin. The yellow square marks the position of the W1M3A observing system, whereas the red circle shows the operational range of the weather radar marked as the red square.

For radionuclides measurements, the underwater gamma-ray spectrometer KATE-RINA [25,44] was exploited. It is a low-resolution gamma-ray spectrometer based on a 3″ × 3″ NaI(Tl) scintillation crystal. It is equipped with all necessary electronic modules in order to provide gamma-ray spectra in a number of channels (512–2048) set by the user. Its low power consumption (less than 1 W after a recent electronics upgrade), operation depth of 400 m and endurance in wide range of operation temperature (−5–50 °C) makes it ideal for prolonged and continuous marine operation in variable and sometimes harsh environmental conditions. On the other hand, its low energy resolution prevents the easy recognition of radionuclides emitting gamma rays of similar energy (e.g., $^{214}$Bi (609 keV) from $^{137}$Cs (669 keV)), although several de-convolution and spectra analyses techniques have been proposed [45] to expand and optimize NaI(Tl)-based detectors in several environmental applications. KATERINA was attached at the submersible part of the buoy at a depth of 6 m near a conductivity temperature depth measurement (CTD) device. It was continuously operating for the whole period of the experiment, acquiring hourly gamma-ray spectra using 1024 channels. The spectra were stored in the internal memory of the acquisition system onboard the buoy and retrieved during maintenance visits to the W1M3A observing system. The spectra analyses output was used to calculate the activity concentration of the observed radionuclides in the seawater in Bq l$^{-1}$. The calibration methodology for quantitative measurements in seawater is based on experimental and simulation results of Full Energy Photo-peak Efficiency produced by the methodology, which is described in detail elsewhere [35]. The calibration methodology of the system was also expanded for quantitative measurements of radionuclides in marine sediment, as described in [33,46]. During the experimental period, co-located measurements of ambient noise were taken by an underwater passive acoustic listening system (UPAL) deployed at a 40 m depth along the hull of the observatory. One of the interesting features of the automated classification and quantification algorithm running in real-time on the UPAL system [47,48] is the identification of the different raindrop bubble size sound generated on the impact of the sea surface. Due to the natural variation of rainfall, the raindrops fall with a terminal velocity in a wide range of sizes and strike the water surface at various

angles of incidence [49]. The distinctive sound generated from the different drop sizes can be described acoustically using an inversion algorithm of the underwater sound [50]. The method is important due to the uncertainty derived from the large inhomogeneity of precipitation in space and time. In addition, the difficulty of measuring these physical parameters over the ocean using conventional methods (i.e., rain gauges and radar) is limited to costal zones and large buoys, such as the one used in this study. Acoustic inversion data are vital for verifying remote sensing measurements of type and intensity of rainfall over the open seas. The physical properties of the bubble size are associated with the impact of the drop on the sea surface, which, in turn, depends on the liquid precipitation drop parameters (i.e., size, shape, angle and speed of splash).

To determine the origin of air masses, establishing a source-receptor relationship, a back trajectory analysis was performed using the HYSPLIT (Hybrid Single-Particle Lagrangian Integrated Trajectory) model [51,52]. The model uses a Lagrangian method to calculate how the trajectory or air parcels move from their initial location since it is based on a moving frame for the estimates of the advection and diffusion. The vertical velocity field from the meteorological data available from the NCEP/NCAR global reanalysis [53] was chosen as the type of vertical motion method used by the model for computation.

## 3. Results

### 3.1. Detected Radionuclides and Dataset Overview

Two representative spectra are presented in Appendix A Figure A1 before and during rainfall on 9 October 2016. During the rainfall, the most intense photo peaks corresponding to radon's $^{222}$Rn progenies $^{214}$Pb (242, 295, 352 keV) and $^{214}$Bi (609, 1120, 1764 keV) were increased, while $^{210}$Pb low-energy gamma rays (46.5 keV) are not able to be detected by the specific spectrometer due to the high Compton background in the related energy region. As concerns radon's $^{220}$Rn progenies, the most intense photo peak of thallium $^{208}$Tl in 2614 keV did not reveal observable increment. The half-life of $^{220}$Rn (55.6 s) is short, and its diffusion into the atmosphere is rather confined in comparison with $^{222}$Rn. Thus, $^{220}$Rn progenies (e.g., $^{212}$Pb, $^{212}$Bi and $^{208}$Tl) accumulation in the atmosphere is much lower, and their fall out (if any) due to rainfalls could not be detected. Moreover, beryllium $^{7}$Be's most intense photo-peak (477 keV) was not detected. The $^{7}$Be concentration in rainwater in a similar latitude, close to the study area, in the period 2014–2016 was measured in the range (0.2–4.1) Bq l$^{-1}$ [13]. However, $^{7}$Be dilution in seawater seems to prevent its detection, as no photo peak can be observed in the corresponding energy region as it is depicted in the subplot of the spectra. Additionally, $^{22}$Na in rainwater activity concentration is in the order of $10^{-3}$ Bq l$^{-1}$ [13], which is far below the minimum detectable activity concentration of the specific spectrometer. Thus, the observed photo peaks of radon $^{222}$Rn progenies $^{214}$Pb, $^{214}$Bi and potassium $^{40}$K were analyzed, and the corresponding activity concentrations were derived [35].

The acquisition lag of the gamma-ray spectrometer was set to 1 h. The other measuring instruments operated on the buoy were set up with different acquisition lags (from 1 min to 1 h), although all were synchronized with the same GPS clock. For the purposes of this work, a database based on hourly synchronized measurements was developed. The database includes atmospheric and meteorological parameters (air temperature, atmospheric pressure, rainfall amount and intensity, wind speed and direction), seawater parameters (surface seawater temperature (SST), evaporation, seawater temperature and salinity at the depth of 6 m) and co-located seawater radioactivity parameters. The latter consists of: (a) the total counting rate (TCR) of each spectrum representing all natural and anthropogenic radionuclides of seawater that emit gamma rays; (b) the activity concentration of radon progenies lead $^{214}$Pb and bismuth $^{214}$Bi and the natural radioisotope of potassium $^{40}$K obtained by the spectral analysis; and (c) the radon progenies ratio $^{214}$Bi/$^{214}$Pb calculated from the activity concentration results. Although the experiment lasted 3004 h, data acquisition problems resulted in a number of data losses that were different for each device; thus, the number (*n*) of mutually available hourly synchronized

data differs in each analysis step, and it is explicitly presented in each diagram. For the parameters obtained with time lag less than 1 h, average values were used. For rainfall height, the cumulative amount (in mm) of rainwater that fell within one hour was considered, whereas for rainfall intensity, the average and maximum values of each rainfall event were used.

### 3.2. Descriptive Analysis of Radioactivity Data during Dry Meteorological Periods

An important task of the work was to evaluate the possibility to observe rainfall events by means of radioactivity parameter variation. For that end, the first step was to investigate the distribution of radioactivity data during dry meteorological periods. In Figure 2, histograms of TCR Figure 2a, $^{214}$Pb Figure 2b, $^{214}$Bi Figure 2c and $^{40}$K Figure 2d are depicted during such periods.

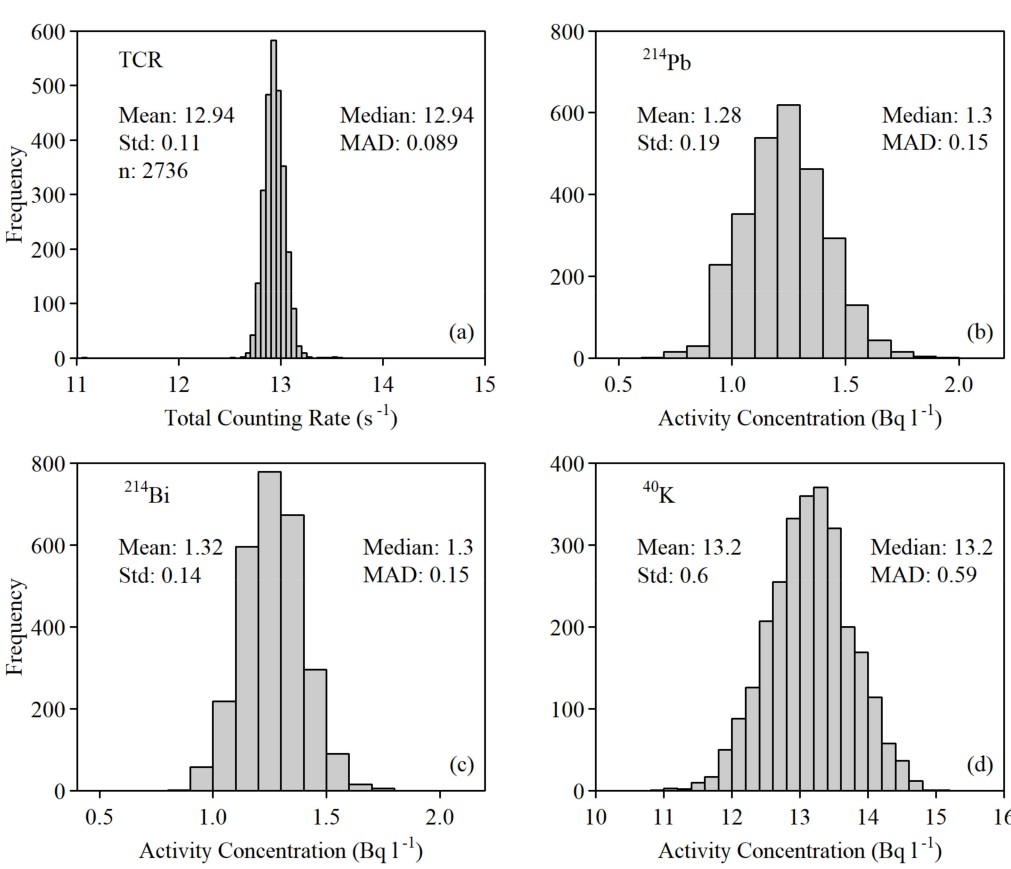

**Figure 2.** Descriptive analysis of TCR (**a**), radon progenies $^{214}$Pb (**b**), $^{214}$Bi (**c**) and $^{40}$K (**d**) activity concentration values during dry meteorological periods.

TCR represents the counting rate of the total number of gamma rays detected during one hour, and it was selected as the indicator for the rainfall event observation purpose due to its low statistical uncertainty. It includes detection events due to all natural and anthropogenic radionuclides present in the seawater that emit gamma rays in a spherical volume around the detector. Although TCR was directly obtained without spectral analysis, its variations are considered to mainly indicate changes of $^{214}$Pb, $^{214}$Bi and $^{40}$K activity concentrations responding to rainfall events since radon progenies and potassium $^{40}$K are the dominant radionuclides in the seawater. The TCR mean and median were found to be equal to 12.94 s$^{-1}$ (with standard (Std) and median absolute deviation (MAD) deviation of 0.11 s$^{-1}$ and 0.15 s$^{-1}$, respectively), so the TCR distribution clearly approximates a normal distribution. Consequently, the TCR mean value standard deviation can be used as the criterion of statistically significant radioactivity variations due to rainfall events. The same analysis is presented in Figure 2b–d for the activity concentration of radon progenies ($^{214}$Pb,

$^{214}$Bi) and $^{40}$K, respectively. As expected, the concentrations of radon progenies are almost equal ($^{214}$Pb: (1.28 ± 0.19) Bq l$^{-1}$, $^{214}$Bi: (1.32 ± 0.14) Bq l$^{-1}$), and its ratio $^{214}$Bi/$^{214}$Pb is close to unity, revealing that during dry meteorological periods, radon progenies in seawater are in the state of radioactive equilibrium with their parent radionuclide of radon $^{222}$Rn. Additionally, the mean value for the $^{40}$K concentration was found to be equal to (13.2 ± 0.6) Bq l$^{-1}$. These values have been considered as background level for radon progenies and $^{40}$K for the rest of the analysis and include radionuclides of seawater during dry meteorological periods and radionuclides that exist in the structural materials of the W1M3A observing system.

### 3.3. Descriptive Analysis of Rainfall Height and Intensity

The number of rainfall events recorded by the rain gauge installed on the top mast of the W1M3A observatory was 71, while it was 46, as reported by the weather radars. Among those, there were 21 radar detections of rainfall events not mutually recorded by the rain gauge, and 45 rainfall records obtained by the rain gauge that were not also detected by the weather radars. The discrepancy is due to the different operation and spatial coverage of rain-gauge and weather radar. As the rain gauge of the buoy was operating exactly upon the site of the underwater experiment and its reliability has been proved in many other studies [54–56], its records were considered more adequate to be involved in further statistical analyses with seawater measurements. In the histograms of Figure 3, the distributions of rain gauge measurements are depicted regarding the cumulative rainfall height over an hour (a), the maximum rainfall intensity (b) and hourly average of rainfall intensity (c). The majority of the rainfall events that occurred during the experiment were very low intensity within the 0.5–1 mm m$^{-1}$ range. More intense rainfalls were recorded after mid-September, in agreement with the climatology of the Ligurian basin, showing periods of drought during spring and summer and intense rainfalls in the fall and winter seasons [57].

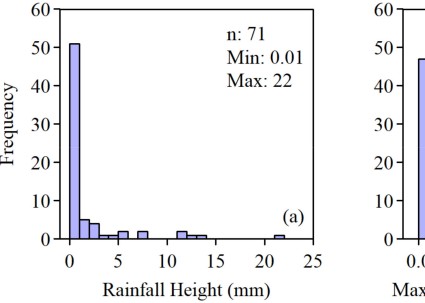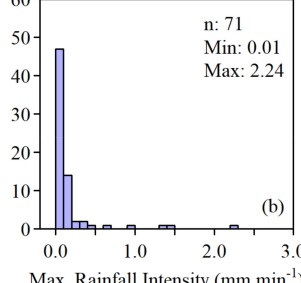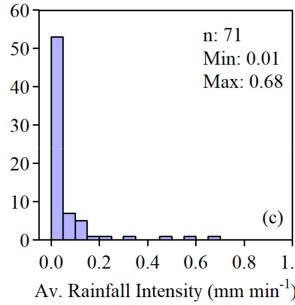

**Figure 3.** Descriptive analysis of rainfall height (**a**), maximum intensity (**b**) and hourly average of intensity (**c**) values, as acquired by the rain gauge of the W1M3A observatory.

### 3.4. TCR Statistical Criterion for Rainfall Recognition

A task of this work was to evaluate the possibility of using TCR as a potential tracer of rainfall events recognition. According to the TCR statistical analysis during dry periods, TCR distribution approximates a normal distribution, so the standard deviation (Std) of the mean value is used to assess the effectiveness of rainfall recognition based on TCR variation during rainfalls. TCR values greater or less than the mean value by 1, 2, and 3 times the Std were examined considering four scenarios (S1–4). The scenarios concern the success (S1) or miss (S2) of rainfall recognition when the rain gauge of the W1M3A observatory recorded rainfall events (rain height greater than 0.1 mm) and the failure (S3) or success (S4) to discriminate rainfall absence when the rain gauge recorded no rainfall (rain height lower than 0.1 mm). The results of the evaluation are presented in Table 1. From a statistical point of view, variations from the mean value greater/less than (at least) two times the Std can be considered significant. Based on that, TCR variations during rainfall were statistically significant in 44% and 38% of the recorded rainfall events according to 2 Std and 3 Std

criteria, respectively. Additionally, during dry meteorological periods, TCR variations were not statistically significant in more than 98% of the measurements. Concerning the 2 Std criterion, the rain height of the rainfall events when TCR did not significantly vary was less than 1 mm in 33 out of 40 times, revealing that TCR effectiveness was lower mainly during rainfalls of low height. To the whole extent of the manuscript, the 2 Std statistical criterion is used, and those events are depicted with red-colored symbols.

**Table 1.** Evaluation using TCR statistical criteria to rainfall recognition according four scenarios **S1–S4**: success (**S1**) or miss (**S2**) of rainfall recognition when the rain gauge recorded rainfall, failure (**S3**) or success (**S4**) to discriminate rainfall absence when the rain gauge recorded no rainfall.

| | **S1. Successful rainfall recognition** | **S2. Fail of rainfall recognition** |
|---|---|---|
| | 1 Std: ($12.83 \text{ s}^{-1} \geq$ TCR $\geq 13.05 \text{ s}^{-1}$) | 1 Std: ($12.83 \text{ s}^{-1} \leq$ TCR $\leq 13.05 \text{ s}^{-1}$) |
| **Rainfalls** | **51 out of 71 (72%)** | **20 out of 71 (28%)** |
| Rain height $\geq$ 0.1 mm | 2 Std: ($12.72 \text{ s}^{-1} \geq$ TCR $\geq 13.16 \text{ s}^{-1}$) | 2 Std: ($12.72 \text{ s}^{-1} \leq$ TCR $\leq 13.16 \text{ s}^{-1}$) |
| (***n* = 71 matchup pairs**) | **31 out of 71 (44%)** | **40 out of 71 (56%)** |
| | 3 Std: ($12.61 \text{ s}^{-1} \geq$ TCR $\geq 13.27 \text{ s}^{-1}$) | 3 Std: ($12.61 \text{ s}^{-1} \leq$ TCR $\leq 13.27 \text{ s}^{-1}$) |
| | **27 out of 71 (38%)** | **44 out of 71 (62%)** |
| | **S3. Faulty recognition of rainfall** | **S4. Success recognition of rainfall absence** |
| **No rainfalls** | 1 Std: ($12.83 \text{ s}^{-1} \geq$ TCR $\geq 13.05 \text{ s}^{-1}$) | 1 Std: ($12.83 \text{ s}^{-1} \leq$ TCR $\leq 13.05 \text{ s}^{-1}$) |
| Rain height < 0.1 mm | **703 out of 2684 (26%)** | **1981 out of 2684 (74%)** |
| (***n* = 2684 matchup pairs**) | 2 Std: ($12.72 \text{ s}^{-1} \geq$ TCR $\geq 13.16 \text{ s}^{-1}$) | 2 Std: ($12.72 \text{ s}^{-1} \leq$ TCR $\leq 13.16 \text{ s}^{-1}$) |
| Excluding 2 h after each rainfall end | **53 out of 2684 (2%)** | **2631 out of 2684 (98%)** |
| as radon progenies decay time | 3 Std: ($12.61 \text{ s}^{-1} \geq$ TCR $\geq 13.27 \text{ s}^{-1}$) | 3 Std: ($12.61 \text{ s}^{-1} \leq$ TCR $\leq 13.27 \text{ s}^{-1}$) |
| | **5 out of 2684 (0.2%)** | **2679 out of 2684 (99.8%)** |

*3.5. Descriptive Analysis of Radioactivity and Salinity Data during Rainfalls*

The distribution of TCR values obtained during rainfalls is depicted in the histogram of Figure 4a. Additionally, the distributions of $^{214}$Pb and $^{214}$Bi activity concentrations are presented in Figure 4b, c, respectively. Comparing with the mean values obtained during dry meteorological periods, most rainfall events resulted in a $^{214}$Bi increase and $^{40}$K decrease. For the case of $^{214}$Pb, almost half of the rainfall events resulted in its increment. During rainfall events when radon progenies and/or $^{40}$K activity concentration was reduced, there is no removal of the radionuclides from seawater. The decrease is attributed to their dilution, which occurs when rainwater containing lower concentrations of radionuclides (than that of background level of seawater) is mixed with the seawater, resulting in an apparent lower activity concentration. Moreover, the distribution of radon progeny ratio $^{214}$Bi/$^{214}$Pb is presented in Figure 4e and salinity records in Figure 4f.

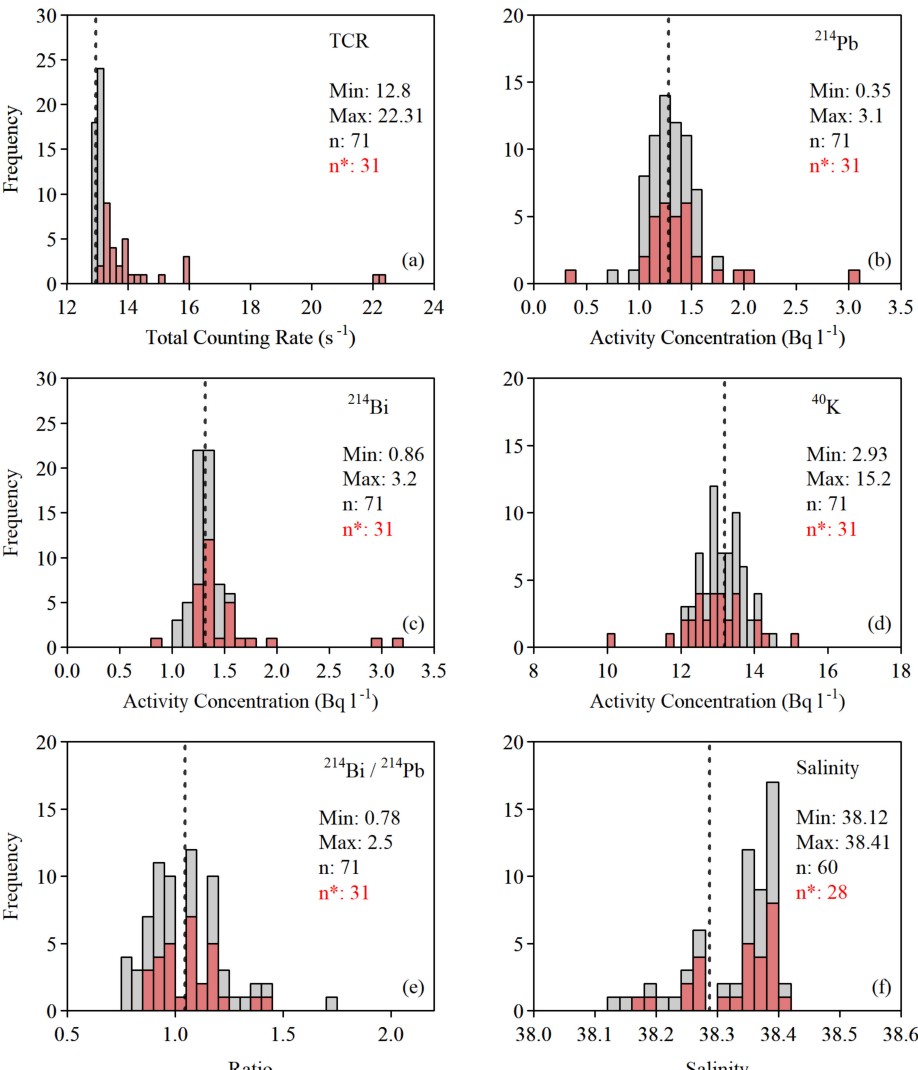

**Figure 4.** The distribution of TCR values (**a**) radon progenies (**b**,**c**) and $^{40}$K (**d**) activity concentration, radon progenies ratio (**e**) and salinity (**f**) during rainfall events (n). When the TCR statistical criterion is fulfilled (n*), the corresponded values are presented with red bars. The mean value during dry meteorological periods of each parameter is depicted by a dotted vertical line.

### 3.6. Raindrop Size Distribution Analysis by Acoustic Method

The technique, as proposed by Nystuen [50], was applied to the measurements acquired in the periods June–August, September and October. Limitations to the technique include the small number of mutual observations of UPAL with the KATERINA system and issues associated with the relative loudness of the largest drops (diameter over 3.5 mm), the relative quietness of the medium drops (diameter 1.2–2.0 mm) and the influence of wind to suppress the signal from the otherwise remarkably loud small drops (diameter 0.8–1.2 mm). Several examples of the sound spectrum recorded during rainfall are presented in Table 2. These are representative of the different "types" of rainfall that were observed during the experiment and illustrate some of the features of the underwater sound that are closely associated with the rainfall drop size distribution (DSD). DSD is highly correlated with the intensity of the impact of the droplet on the sea surface, indicating different splashing characteristics. These features are consistent with laboratory studies of sound generated by individual raindrops [58,59]. Almost 60% of the records are attributed to raindrops of the small-size class of diameters in the range (0.8–1.2) mm.

**Table 2.** The physics of the drop splash associated with different types of drop size.

| Drop Size | Diameter (mm) | Sound Source | Frequency Range (kHz) | Splash Character | Percentage of Presence (%) |
|---|---|---|---|---|---|
| Tiny | <0.8 | Silent | | Gentle | - |
| Small | 0.8–1.2 | Loud bubble | 13–15 | Gentle, with bubble every splash | 59.94 |
| Medium | 1.2–2.0 | Weak impact | 1–30 | Gentle, no bubbles | 38.36 |
| Large | 2.0–3.5 | Loud impact bubbles | 1–35 2–35 | Turbulent irregular bubble entrainment | 1.65 |
| Very large | >3.5 | Loud impact Loud bubbles | 1–50 1–50 | Turbulent irregular bubble entrainment penetrating jet | 0.05 |

### 3.7. Time Series

The presentation of all data in the form of a time series is not necessary as there was an extended period without rainfall events (e.g., during July–August when only three rainfall events took place). Instead, in Figures A2–A5 of Appendix A, indicative periods are presented regarding the most representative rainfall events of June, August, September and October of 2016, respectively. Observing the time series, the following outcomes can be deduced:

- There is not an obvious association between rainfall height and TCR values or any of radionuclides activity concentration. Rainfall events of very different heights result in very strong TCR increment. For example, in August (Figure A3), rainfall events with a height difference higher than two orders of magnitude (0.3 and 12 mm) resulted in high TCR values of 14 and 15.5 $s^{-1}$, respectively.
- When TCR variation is statistically significant, there is a delay from 2 to 3 h occurs after the end of a rainfall before TCR values return to their pre-rainfall values. The same is also true for radon progenies. The most representative example was observed in October (Figure A5). The delay of TCR is explained by the time needed for radon progenies contained in rainwater to totally decay (3–5 times their half-lives which equal to 75–125 min) and reach their background levels.
- In general, TCR immediately responded in all cases when the statistical TCR criteria were fulfilled, and it was proved as a reliable tracer for rainfall observations. Although TCR variation is mainly ascribable to radon progenies fall out, the variation of radon progenies activity concentration sometimes was too weak to be observed (e.g., Figures A2 and A3). However, even a slight enrichment of radon progenies in seawater results in significant variation of TCR. This is due to plenty of gamma rays corresponding to radon progenies (especially $^{214}$Bi) that contribute to the whole spectrum and significantly increase the value of TCR.

## 4. Discussion

### 4.1. Cloud Origin Impact in Radioactivity Increment during Rainfalls

An important factor that may affect the increase of seawater radioactivity during rainfalls is the richness of rainwater in radon progenies and consequently the origin of the rain clouds. Clouds formed upon terrestrial areas are expected to be richer in radon progenies than clouds formed upon seas. As an indicator of cloud origin and the transportation path, the wind direction recorded 10 m above the sea surface by the meteorological station of the W1M3A observing system is used. The wind direction may be different in the altitudes of cloud formation and, in general, the meteorological systems have rather complicated trajectories; however, it can be used as an indicator of TCR variation dependency on air masses' transportation direction.

In the polar plots of Figure 5, TCR values (radial axes) are plotted with respect to wind direction without rainfalls (a) and during rainfall events (b). As the exact values of wind direction (in degrees) vary over an hour, they have been replaced by broader orientation points. During dry meteorological conditions, TCR values show no dependency on the wind direction as the values are evenly distributed along all wind directions. However,

during rainfall events of statistically significant TCR variation (red circular points), the most values reveal angular dependency.

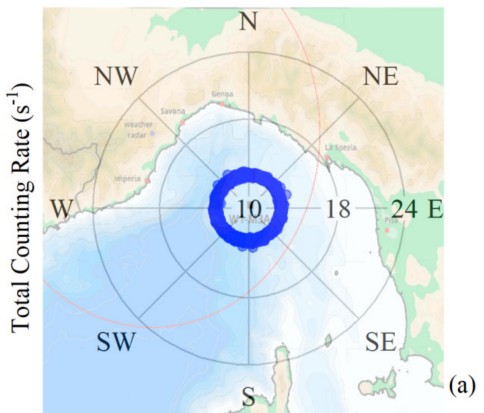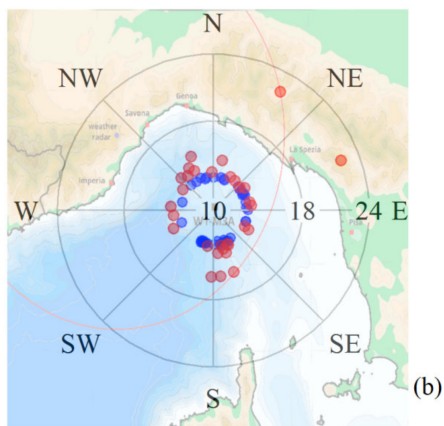

**Figure 5.** Analysis of TCR values with wind direction during dry meteorological conditions (**a**) and rainfalls (**b**). The events of statistically significant variation of TCR are presented with red dots.

The analysis was repeated for the case of radon daughters and their ratio $^{214}$Bi/$^{214}$Pb and is presented in the Figure 6. The activity concentration of both $^{214}$Pb and $^{214}$Bi in seawater follows the dependency of TCR with the wind direction, strengthening the outcome that a TCR increase during rainfalls is mostly induced by rainfall events rich in radon progenies. Unlike the cases of TCR and radon progenies, there is no obvious relation between the radon ratio and the wind direction. Analogous results have been reported from other geographical regions [60–62].

To further investigate the role of aerial mass pathways in a TCR increase during rainfall, their back-trajectories were computed for the W1M3A mooring site by the HYSPLIT model with 2 h intervals, starting at the time of the rainfall event and a 48 h pathway at 0 m above the ground level (AGL). The obtained back-trajectories and the corresponding maximum value of TCR are depicted in the Figure 7 for the representative rainfall events of June, September and October presented in Appendix A Figures A2–A5. In each diagram, the different colored lines regard the 6 back-trajectories initiated in a period of 0–10 h before the rainfall starts with a time step of 2 h.

The model outputs evidence airflows confined in a very narrow layer (less than 500 m height above the ground level) and pathways over two days limited to about 300 km. During the rainfall events of June and August (Figure 7a–c), air masses were coming from the French coast, and the trajectories were mainly expanded over the sea. In those cases, the TCR value was moderately increased. During the rainfall events of September and October (Figure 7d–f), the origin of the pathways steadily turns to the mainland of Italy and Corsica Island, and the trajectories were expanded longer over terrestrial areas. In those cases, the TCR value was further increased and during the rainfall event of 8–9 October (Figure 7f), when the majority of the trajectories were over the mainland of Italy, TCR values reached the maximum value of 22.3 s$^{-1}$. Although further investigation may reveal the favorable combination of cloud origin and heights for the TCR increment, its dependency from the aerial mass origin and pathways was clearly observed.

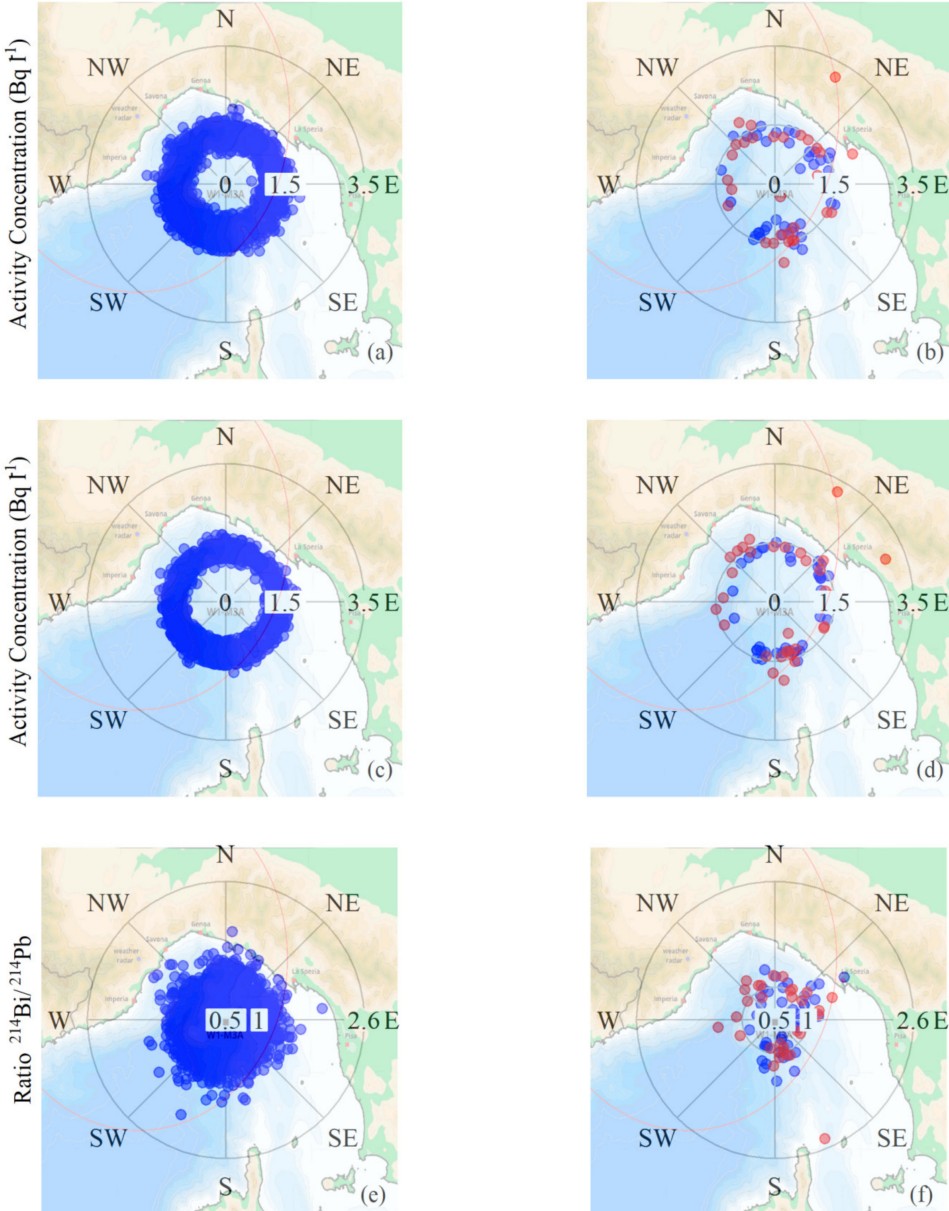

**Figure 6.** Wind direction analysis of activity concentration values of $^{214}$Pb (**a**,**b**), $^{214}$Bi (**c**,**d**) and the ratio $^{214}$Bi/$^{214}$Pb (**e**,**f**), during dry meteorological conditions in the first column and during rainfalls in the second. The events of statistically significant variation of TCR are presented with red dots.

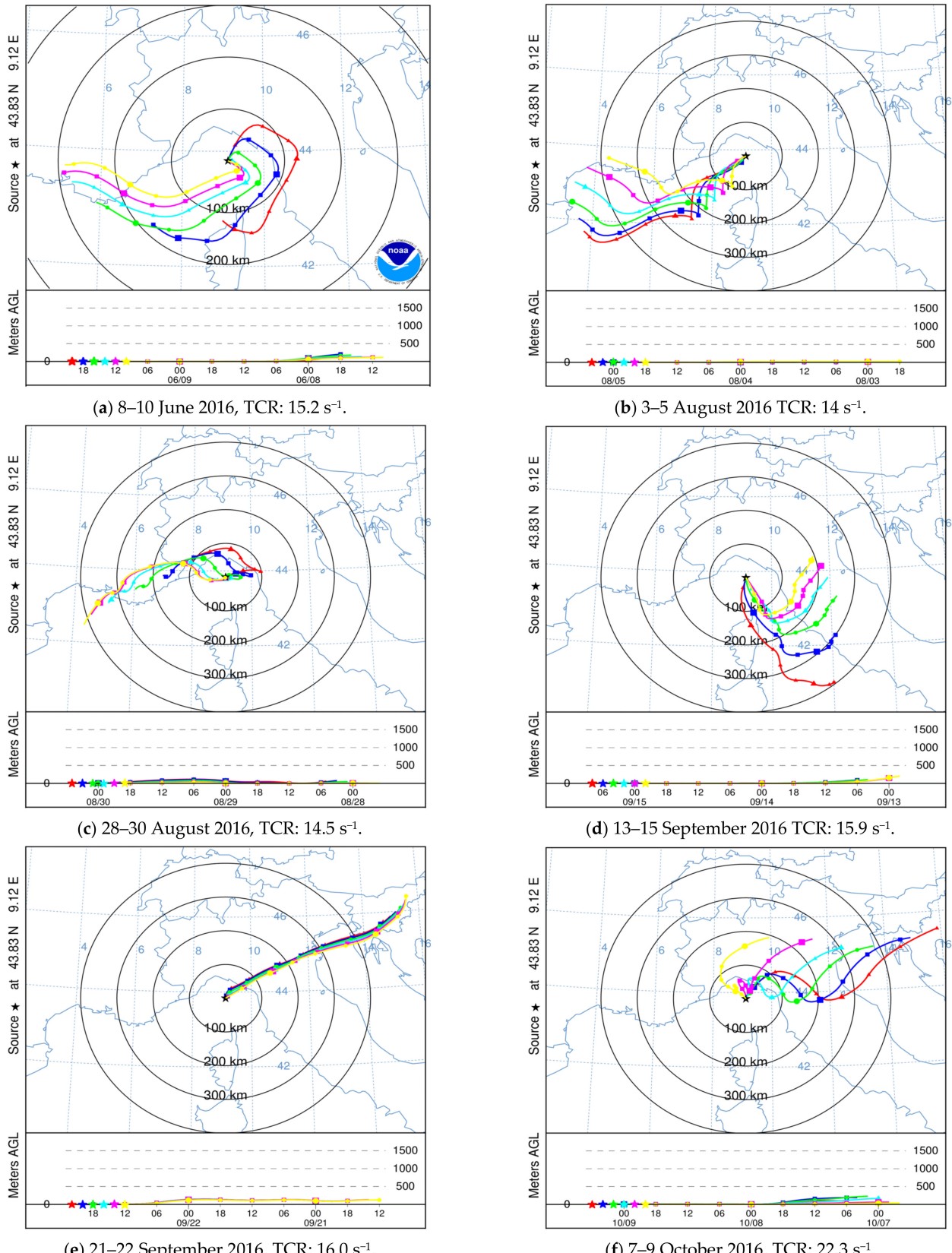

(**a**) 8–10 June 2016, TCR: 15.2 s⁻¹.

(**b**) 3–5 August 2016 TCR: 14 s⁻¹.

(**c**) 28–30 August 2016, TCR: 14.5 s⁻¹.

(**d**) 13–15 September 2016 TCR: 15.9 s⁻¹.

(**e**) 21–22 September 2016, TCR: 16.0 s⁻¹.

(**f**) 7–9 October 2016, TCR: 22.3 s⁻¹.

**Figure 7.** Back-trajectories obtained with the HYSPLIT model using NCEP/NCAR global reanalysis meteorological fields computed at 0 m above the ground level (AGL) and runtime of 48 h for rainy events occurred in June, August, September and October 2016. Different colors indicate 48 h periods starting 0 h (red), 2 h (blue), 4 h (green), 6 h (cyan), 8 h (magenta) and 10 h (yellow) before the rain event.

### 4.2. Association of Radioactivity Data with Rainfall Parameters

In the Figures 8 and 9, the associations of hourly obtained rainfall height (Figure 8) and average rainfall intensity (Figure 9) with TCR, radon progenies and $^{40}$K activity concentration and the ratio $^{214}$Bi/$^{214}$Pb are depicted.

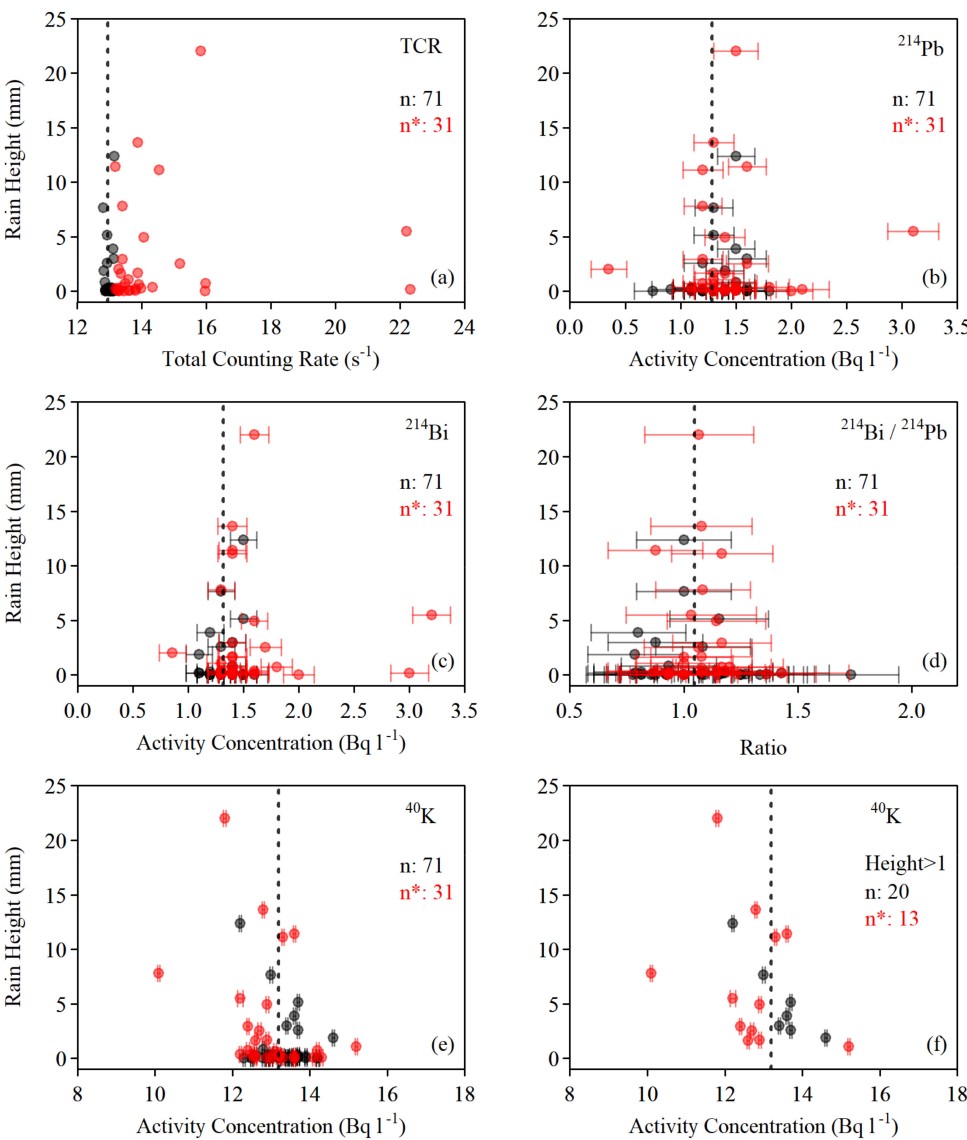

**Figure 8.** The rainfall height over TCR (**a**), $^{214}$Pb (**b**), $^{214}$Bi (**c**), radon progenies ratio (**d**), $^{40}$K (**e**) and $^{40}$K (**f**) for rain heights higher than 1 mm. The total number of events is n, and the events of statistically significant variation of TCR (n*) are presented with red dots. The mean value of each parameter during dry meteorological periods is presented with dotted line.

Previous studies in the atmosphere have revealed a decreasing, non-linear relation of rainfall intensity with radon progenies, and the ratio $^{214}$Bi/$^{214}$Pb has been used for rainwater aging [6,9,17,19]. Thus, the investigation of potential associations between radionuclides in seawater with rainfall parameters is of great interest, although the underwater detection of radon progenies in rainwater by means of gamma-ray spectroscopy significantly differs from atmospheric detection. The most important difference concerns the inability to measure radon progenies solely in rainwater. In the seawater volume above and around the gamma-ray spectrometer, rainwater (potentially rich in radon progenies without $^{40}$K) mixes with seawater (rich in $^{40}$K), so the underwater measurement of radionuclides regards the mixed rain/seawater mass. Subsequently, the rain height represents the amount of

rainwater that is cumulatively added triggering and retaining the mixing process, while the rainwater intensity represents the rapidness of the mixing.

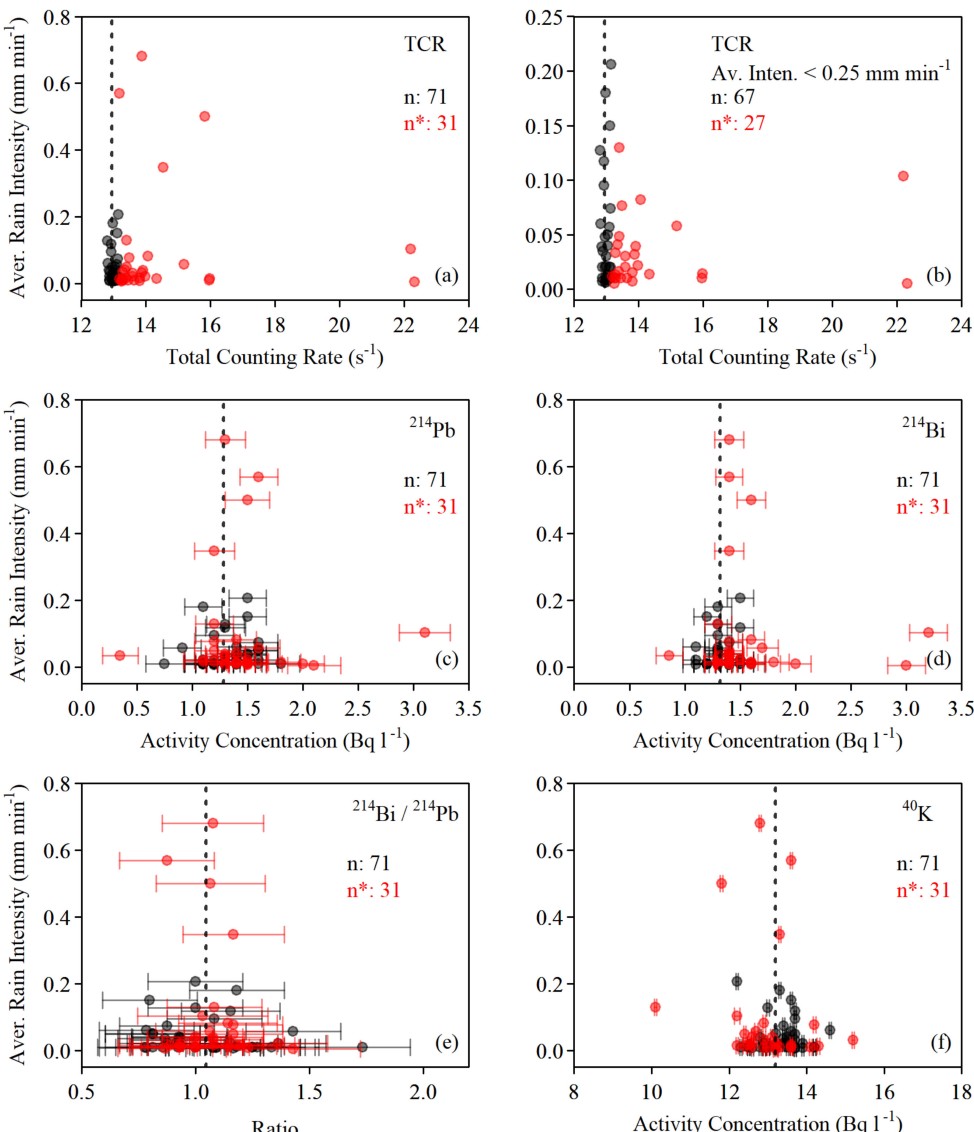

**Figure 9.** The average rainfall intensity over TCR (**a**), TCR for intensity lower than 0.25 mm min$^{-1}$ (**b**), $^{214}$Pb (**c**), $^{214}$Bi (**d**), radon progenies ratio (**e**) and $^{40}$K (**f**). The total number of events is *n*. The events of statistically significant variation of TCR (*n\**) are presented with red dots. The mean value of each parameter during dry meteorological periods is presented with a dotted line.

The most notable outcomes regard:

- The rain height exhibits an inverse non-linear relation with TCR, as presented in Figure 8a, and almost the same behavior with radon progenies (b) and (c).
- Rain heights greater than 5 mm merely affect the activity concentration of radon progenies, revealing that the amount of rainwater is not the major factor that affects the concentration of radon progenies
- As can be observed in Figure 8d, there are very few rainfalls of very low height (<1 mm) leading to an observable change of radon progenies ratio $^{214}$Bi/$^{214}$Pb. Also, considering its high uncertainty, the ratio was not found related with rainfall height.
- Although there is not an obvious relation of $^{40}$K with the rainfall height during weak rainfalls (Figure 8e), for events of height higher than 1 mm, an inverse relation can be observed (Figure 8f). As a result, the activity concentration of $^{40}$K exhibits a

promising tracer of the rainfall height, although the sample size was not enough to apply regression analysis. A possible interpretation regards the dilution of $^{40}$K in the mixed rain–seawater volume. As rainwater contains negligible quantities of salts, during rainfalls, all seawater salts, such as potassium and its radioisotope $^{40}$K, are temporarily diluted. Higher amounts cause a higher degree of dilution, leading to an apparently lower $^{40}$K activity concentration. The effect can be further studied, leading to an indirect rainwater quantification based on $^{40}$K measurements.

- The average rainfall intensity exhibits an inverse non-linear relation with TCR, as presented in Figure 9a. The relation is more evident in the case of low-intensity (<0.25 mm min$^{-1}$) rainfalls that cause statistically significant TCR variation (red dots), as presented in Figure 9b. The same behavior was also observed relating to the maximum intensity of each rainfall event with TCR (figure not included to avoid repetitiveness).

- An analogous inverse non-linear relation can be observed for the case of $^{214}$Bi when the rainfalls cause statistically significant TCR variation (red dots), as presented in Figure 9d. Additionally, $^{214}$Pb seems to be very weakly related with the rainfall intensity, as can be observed in Figure 9c. During the period of the experiment, low-intensity rainfall events seem to be related with rainwater of higher radon progenies concentrations.

- Very few events of low intensity led to an observable change in the radon progenies ratio $^{214}$Bi/$^{214}$Pb, as presented in Figure 9e.

- The activity concentration of $^{40}$K does not reveal an overt relation with the average rain intensity. The same is also true for the case of the maximum intensity (figure not included).

### 4.3. $^{40}$K Associations with Radon Progenies and Salinity during Rainfalls

As outlined in the previous section, during rainfalls, rainwater potentially rich in radon progenies mixes with rich-in-$^{40}$K seawater. In Figure 10, the relation of $^{40}$K with $^{214}$Pb (Figure 10a), $^{214}$Bi (Figure 10b), the ratio $^{214}$Bi/$^{214}$Pb (Figure 10c) and seawater salinity (Figure 10d) at the depth of 6 m are presented. The most notable outcomes regard:

- There is no overt relation between $^{40}$K with $^{214}$Pb; however, almost a linear invert relation is observed between $^{40}$K and $^{214}$Bi. Although the amount of available data is not sufficient for statistically precise regression, the observed relation can be used in future application, aiming for indirect estimation of rainwater height and/or intensity. As can be observed in Figure 8f, a lower $^{40}$K activity concentration is related with a greater amount of rainwater and higher $^{214}$Bi. This is the first piece of evidence that the relation between $^{40}$K and $^{214}$Bi may be used to investigate the mixing process between rain- and seawater. The mixing process can be associated with the rainwater amount and/or rainfall intensity, providing an additional indicator in precipitation studies.

- An analogous but much weaker relation between $^{40}$K and radon progenies ratio $^{214}$Pb/$^{214}$Pb can be observed in Figure 10c.

- In previous work in Mediterranean Sea [63], a linear positive relation of the $^{40}$K activity concentration with seawater salinity was observed, during dry meteorological periods, concerning the fact that more salty seawater contains higher concentrations of salts and consequently $^{40}$K. Additionally, an indirect way of seawater salinity measurements based on $^{40}$K was proposed. In this work, the expected linearity cannot be observed considering all of the data. The rainfall events of significant TCR variations (presented by the red points in the Figure 10d) do reveal an almost linear trend between $^{40}$K and salinity, as expected; however, considering the data of no significant statistical variation of TCR, no relation can be observed. The discrepancy can be attributed to the different measuring methods of $^{40}$K and salinity. Salinity measurements are realized by an automated sampling procedure of small quantities of seawater exactly at the deployment point of the CTD instrument, so salinity measurements are rather

point measurements. On the contrary, using in situ gamma-ray spectrometry, the detection of gamma rays emitted by $^{40}$K is realized in a spherical volume (effective volume) of 10–15 m$^3$ around the deployment point [35]. When the amount of rainwater was high, the dilution of $^{40}$K from the surface down to the deployment depth (and inside the whole of the effective volume) tended to be more homogenized, and the measurements of the two instruments followed a linear trend. On the other hand, during events with a low amount of rainwater, the dilution of $^{40}$K mainly took place close to the surface, and the concentration of $^{40}$K was not homogenized in the whole of the effective volume. As a result, the two instruments seem to measure water masses of different degrees of $^{40}$K homogeneity, so the obtained results do not follow the expected linearity.

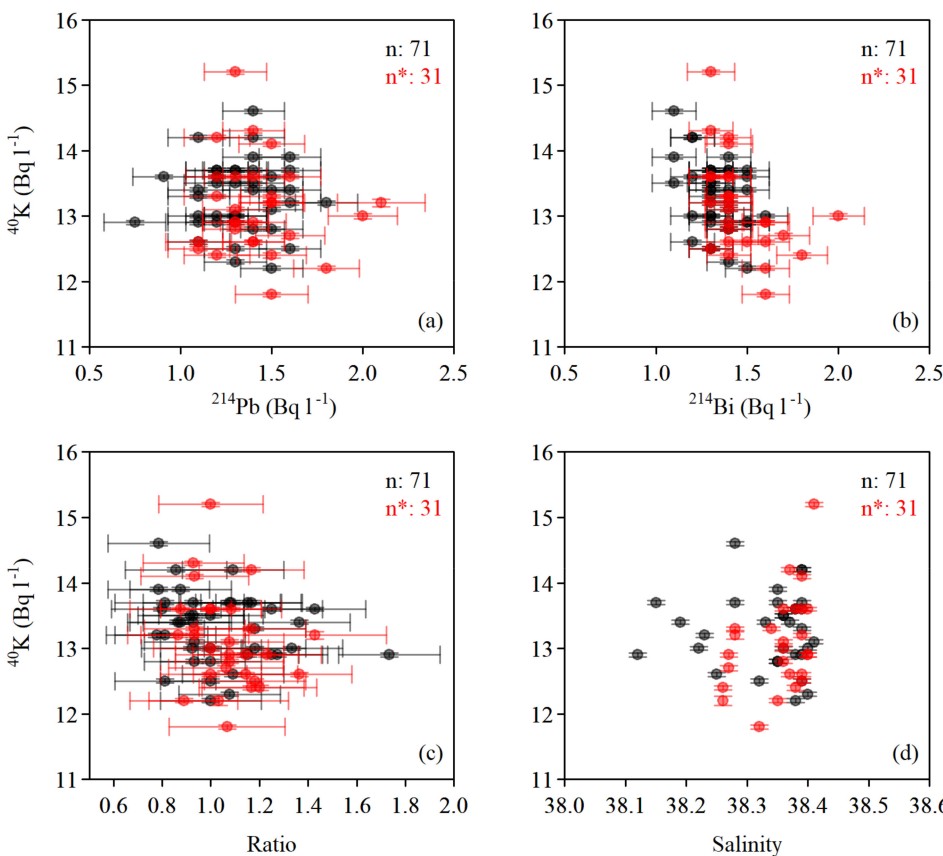

**Figure 10.** $^{40}$K over radon progenies $^{214}$Pb (**a**), $^{214}$Bi (**b**), their ratio (**c**) and salinity (**d**) at the depth of 6 m during rainfalls.

### 4.4. Association of Radioactivity and Rain Drop Size Distribution

During the period of the experiment, the dominant raindrop size class was considered small with a raindrop diameter in the range of 0.8–1.2 mm. The number of raindrops of that particular size bin of each rainfall event is represented by the DSD. Therefore, as a final task, any possible association between radioactivity data and the rain drop size distribution obtained by the acoustic method was attempted. The number of mutual measurements was low ($n$ = 24), and a TCR statistically significant variation was observed in only $n$* = 12 cases, so it was not possible to deduce statistically significant valid results. However, as for preliminary results, the obtained data are presented in Figure A6 of Appendix A where TCR versus DSD are medium correlated with low biased. In this case, the two high DSD values are not taken into consideration. Although those preliminary data are not sufficient for further statistical analysis, they can be considered promising, and data of future application may reveal the potential association of drop counts and radioactivity.

## 5. Conclusions

The main advantage of investigating precipitation events under the sea using natural radionuclides as tracers is the possibility to obtain quantitative results of their activity concentration. In the atmosphere, the quantification is very difficult by means of in situ gamma-ray spectrometry as the effective volume is enormous. In the marine environment, the effective volume is considerably smaller due to the attenuation of gamma rays within the seawater. In this work, activity concentrations of radon progenies and $^{40}$K, based on recent quantification methodologies, were obtained with a time lag of one hour. Although the statistical needs of a profound analysis demand prolonged acquisition periods covering all seasons of a year, the following specific conclusions were deduced from this work:

- TCR has been proven as a good indicator to identify rainfall events. It increases during rainfall events and exhibits a statistically significant increase during 33 out of 71 events, according to the statistical criterion of 2 Std.
- The TCR increment during rainfalls is attributed to radon progenies of $^{214}$Pb and $^{214}$Bi contained in rainwater with the latter being the major contributor to TCR variations due to its higher number of emitted gamma rays.
- TCR values and radon progenies activity concentrations increase during rainfalls and remain higher than the background level for a time of 2–3 h after the events due to the radon decay process.
- The marine environment introduces $^{40}$K as another (along with radon progenies) promising radio tracer, as its dilution during rainfalls was associated with the rainfall height.
- TCR and radon progenies revealed an increasing non-linear trend with both rainfall height and intensity. However, more results are needed to estimate a function that would approach their relation.
- Cloud origin significantly affects TCR and radon progenies increment during rainfalls. Aerial mass trajectories extended upon terrestrial areas of Italy's mainland resulted in the highest increment.

For future marine applications of precipitation investigation using radionuclides as tracers the following aspects should be considered:

- Gamma-ray spectra obtained with a time lag of 1 h during rainfall provide results of good statistics; however, a shorter acquisition time of 20–30 min is suggested to provide more detailed data during rainfall events with duration longer than 1 h. Additionally, a shorter acquisition time is suggested in order to more effectively study shorter than 1 h events.
- A deployment depth of 6 m effectively reduced the contribution of cosmic rays and atmospheric radionuclides to the acquired spectra. However, precipitation events poor in radon progenies did not induce statistically significant variations. A shallower deployment depth with a maximum depth of 3 m is suggested for more effective coverage of the surficial seawater layer from the effective volume of the gamma-ray spectrometer.
- Underwater in situ gamma-ray spectrometers of medium to medium-high resolution may provide more radionuclides as potential tracers (e.g., $^{208}$Tl, $^{137}$Cs).

This work intends to highlight the possibility of natural radionuclides exploitation to reveal new research opportunities in water cycle and climate research. The recent technological achievements described in this work provide the capacity of a global network of remote and continuous monitoring of marine radioactivity, augmenting the use of radio-tracers in multipurpose climate and radiological studies.

**Author Contributions:** Conceptualization, D.L.P. and S.P.; Data curation, D.L.P. and S.A.; Formal analysis, D.L.P.; Investigation, D.L.P.; Methodology, D.L.P., S.P., C.T. and E.G.A.; Resources, S.P., C.T. and M.N.A.; Software, S.P., R.B., M.N.A. and S.A.; Supervision, D.L.P.; Validation, D.L.P., C.T. and R.B.; Visualization, D.L.P. and S.P.; Writing—original draft, D.L.P. and S.P.; Writing—review and

editing, D.L.P., C.T., R.B. and E.G.A. All authors have read and agreed to the published version of the manuscript.

**Funding:** This work has received funding from the European Union Seventh Framework Programme (FP7/2007-2013) under grant agreement n° [312463], [FixO3].

**Institutional Review Board Statement:** Not applicable.

**Informed Consent Statement:** Not applicable.

**Data Availability Statement:** Acoustic and radioactivity data (June 2016–October 2016) are available at the following address: http://www.w1m3a.cnr.it/OI1/modules/site_pages/fixo3_TNA.php, (accessed on 23 July 2021). Meteorological data used for the computation of backward trajectories were freely available from the National Centers for Environmental Prediction/National Weather Service/NOAA/U.S. Department of Commerce. 1994, updated monthly. NCEP/NCAR Global Reanalysis Products, 1948–continuing. Research Data Archive at https://psl.noaa.gov/data/gridded/data.unified.html, (accessed on 23 July 2021). The meteorological data used for the analysis are freely available from the portal of the Copernicus Marine Environment Monitoring Service ( http://marine.copernicus.eu, (accessed on 23 July 2021)).

**Acknowledgments:** The authors gratefully acknowledge the NOAA Air Resources Laboratory (ARL) for the provision of the HYSPLIT transport and dispersion model and/or READY website (https://www.ready.noaa.gov, (accessed on 23 July 2021)) and the contribution of the Agenzia Regionale per la Protezione Ambientale Liguria (ARPAL) that jointly operates the radar system with ARPA Piemonte for providing part of the Monte Settepani data. The work is dedicated to our admired colleague Jeffrey Aaron Nystuen (1957–2020).

**Conflicts of Interest:** The authors declare no conflict of interest.

## Appendix A

Representative gamma-ray spectra were obtained before and during a rainfall of 9 October 2016. The red line regards a spectrum obtained one hour before the rainfall initialization and the blue line the one obtained during the first hour of the rainfall. The energies of the most intense photo-peaks corresponding to $^{214}$Pb (242, 295, 352) keV, $^{214}$Bi (609, 1120, 1764) keV and $^{40}$K (1461 keV) are depicted. In the inset subplot, the energy range 200–700 keV has been focused on in order to assess the possibility of $^{7}$Be photo-peak in 477 keV presence. $^{7}$Be photo-peak cannot be observed. The increase of the Compton background is mainly attributed to radon $^{222}$Rn progenies increment.

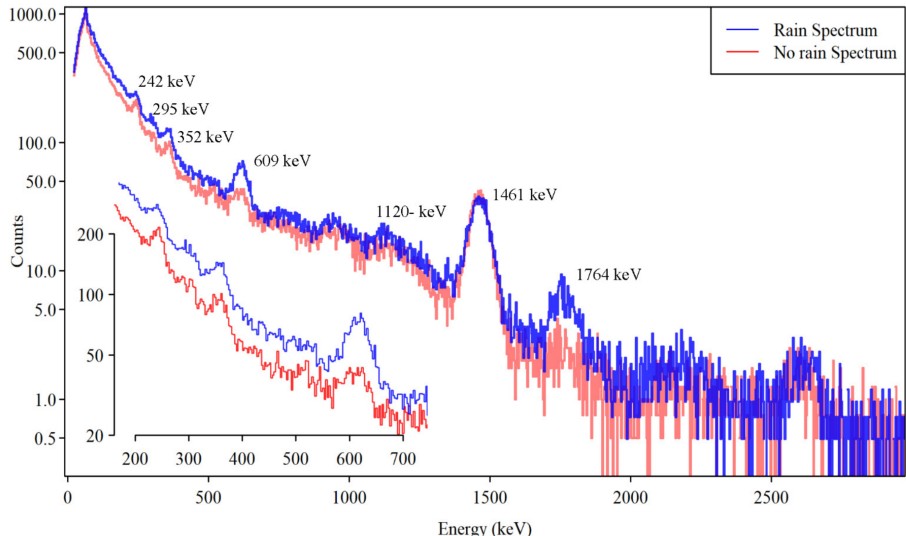

**Figure A1.** Representative gamma-ray spectra obtained before (red line) and during (blue line) the rainfall events of 9 October 2016. On the inset subplot, the energy range 150–750 keV is magnified to show the absence of $^{7}$Be 477 keV photo-peak from the spectrum during rainfalls.

Time series of radioactivity data with rainfall height during the most representative rainfall events of June (Figure A2), August (Figure A3), September (Figure A4) and October (Figure A5) 2016, respectively.

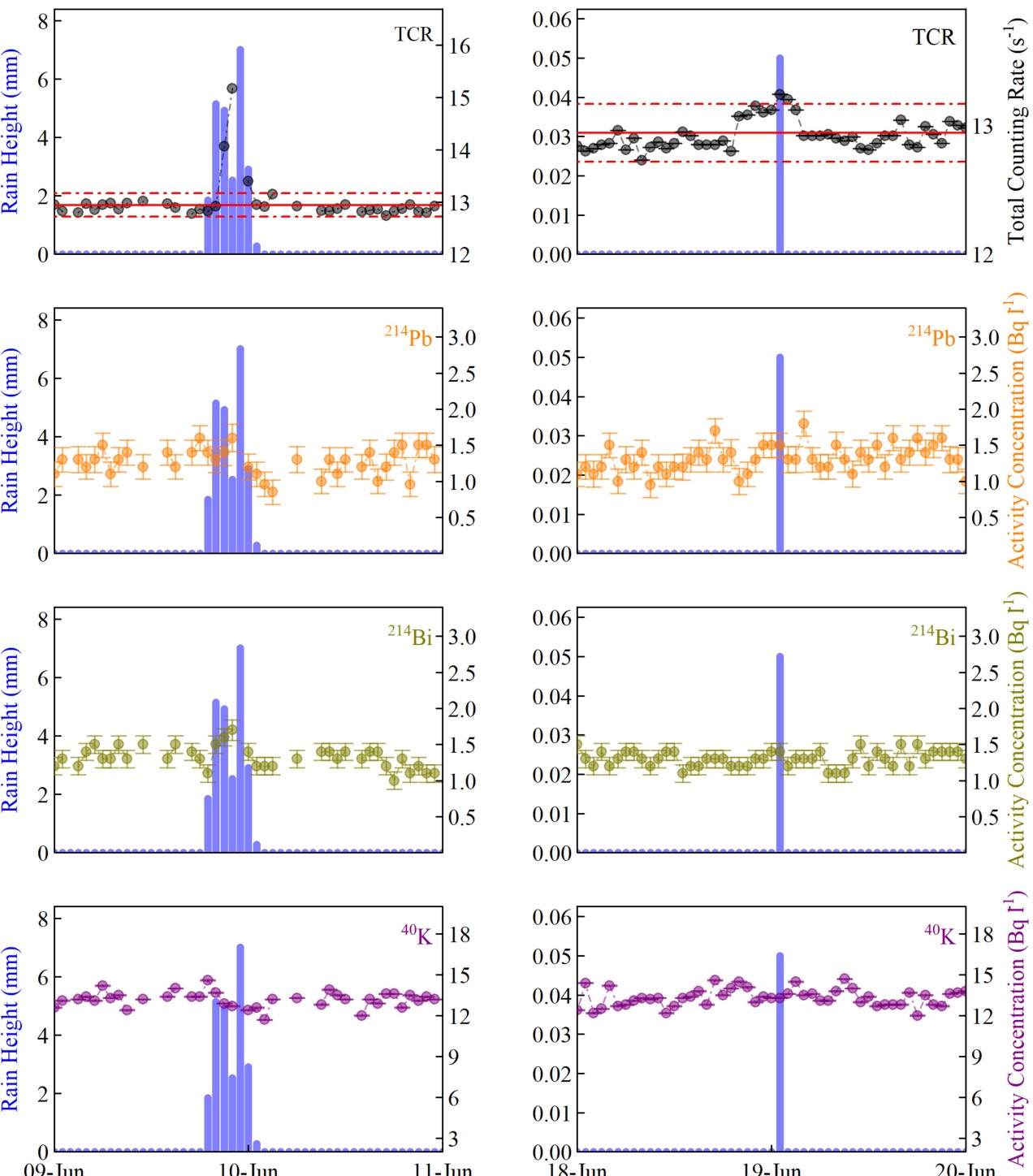

**Figure A2.** Representative rainfall events of June 2016. The *y*-axis of the first column regards rain height values (blue bars) while of the second TCR values, $^{214}$Pb, $^{214}$Bi and $^{40}$K activity concentration (circular points). In the subplots of TCR, the mean value during dry meteorological periods is depicted as a red solid line, and the range of the statistically significant criterion is depicted with red dotted lines.

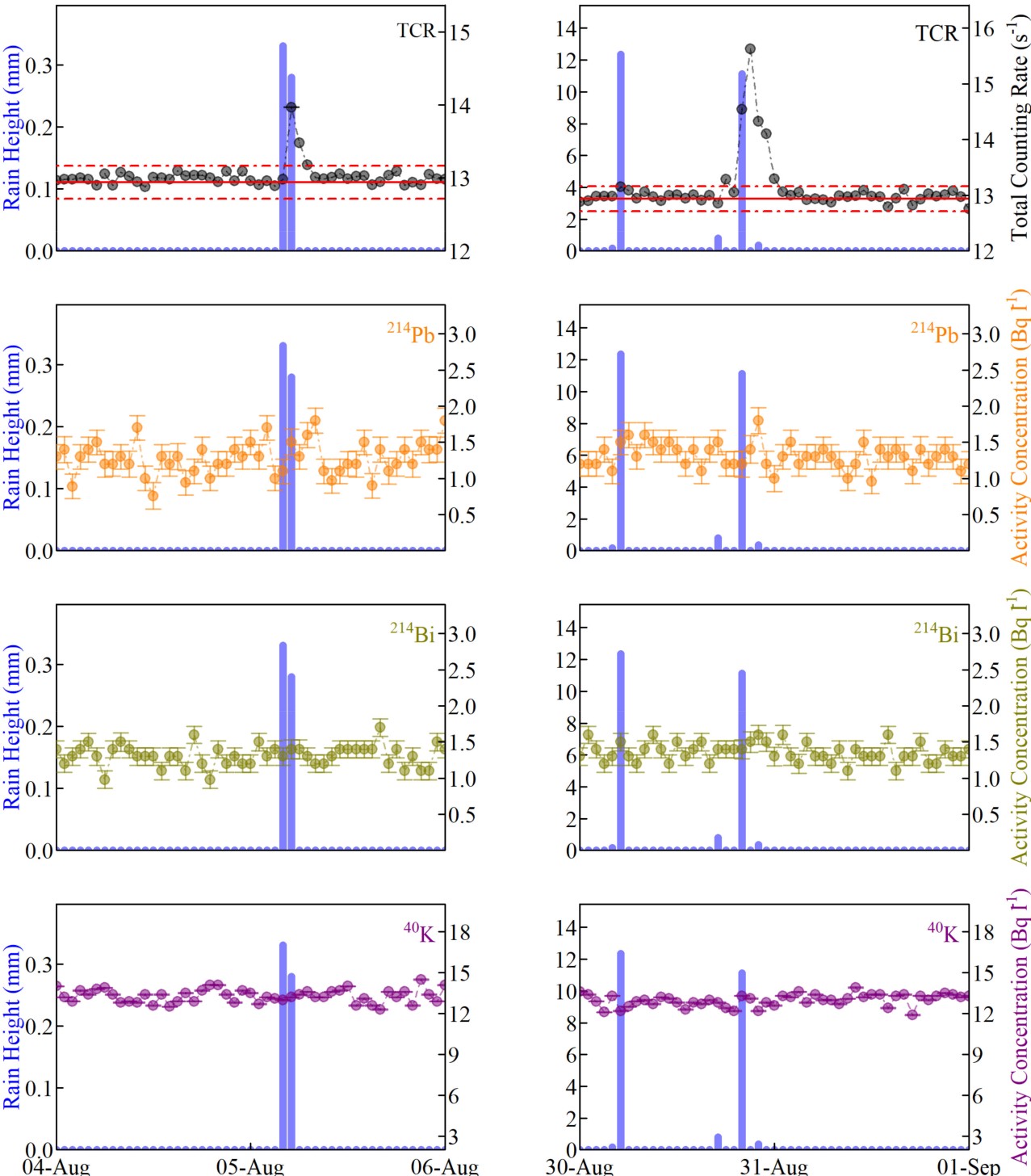

**Figure A3.** Representative rainfall events of August 2016. The *y*-axis of the first column regards rain height values (blue bars) while of the second TCR values, $^{214}Pb$, $^{214}Bi$ and $^{40}K$ activity concentration (circular points). In the subplots of TCR, the mean value during dry meteorological periods is depicted as a red solid line, and the range of the statistically significant criterion is depicted with red dotted lines.

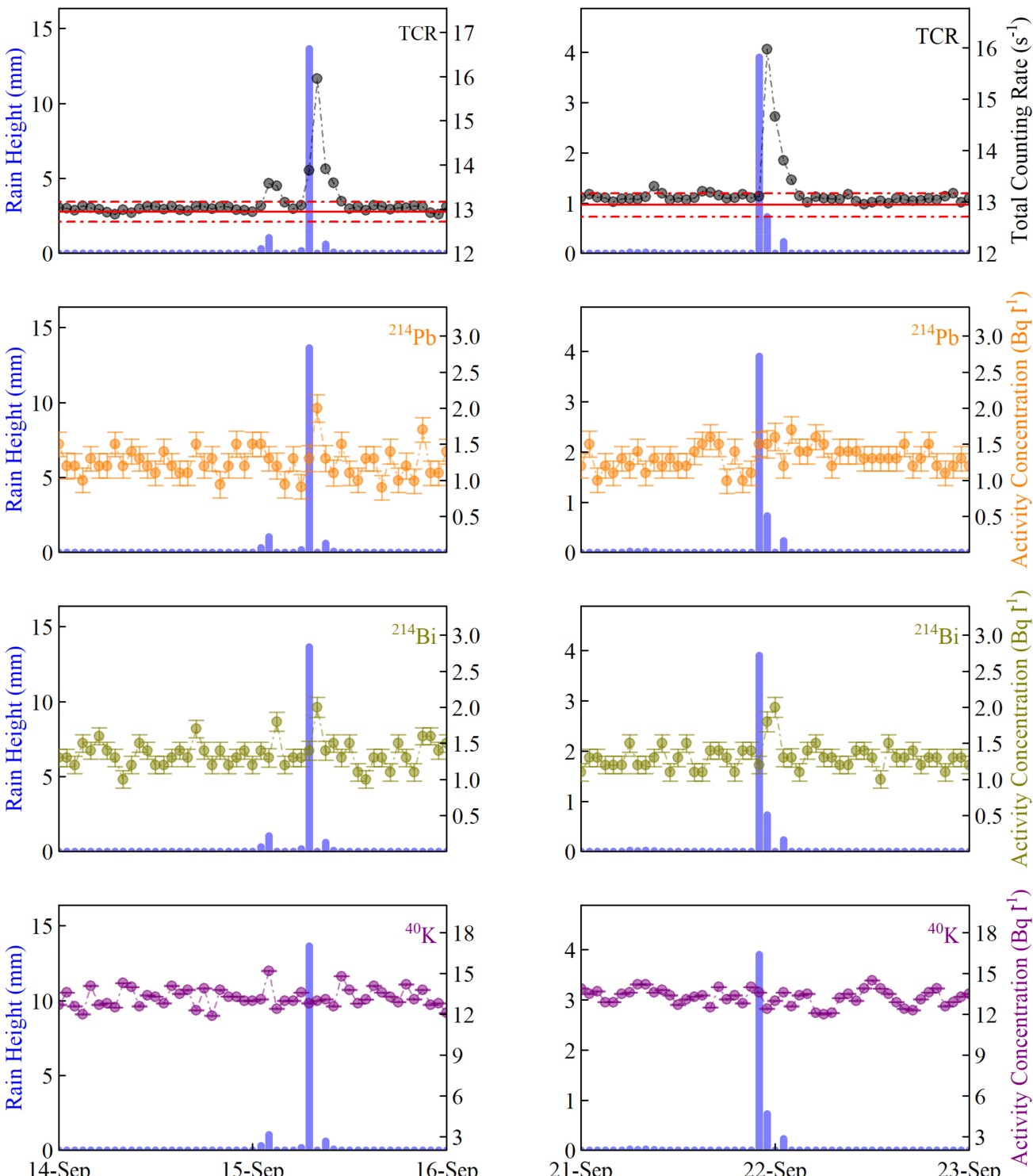

**Figure A4.** Representative rainfall events of September 2016. The *y*-axis of the first column regards rain height values (blue bars) while of the second TCR values, $^{214}$Pb, $^{214}$Bi and $^{40}$K activity concentration (circular points). In the subplots of TCR, the mean value during dry meteorological periods is depicted as a red solid line, and the range of the statistically significant criterion is depicted with red dotted lines.

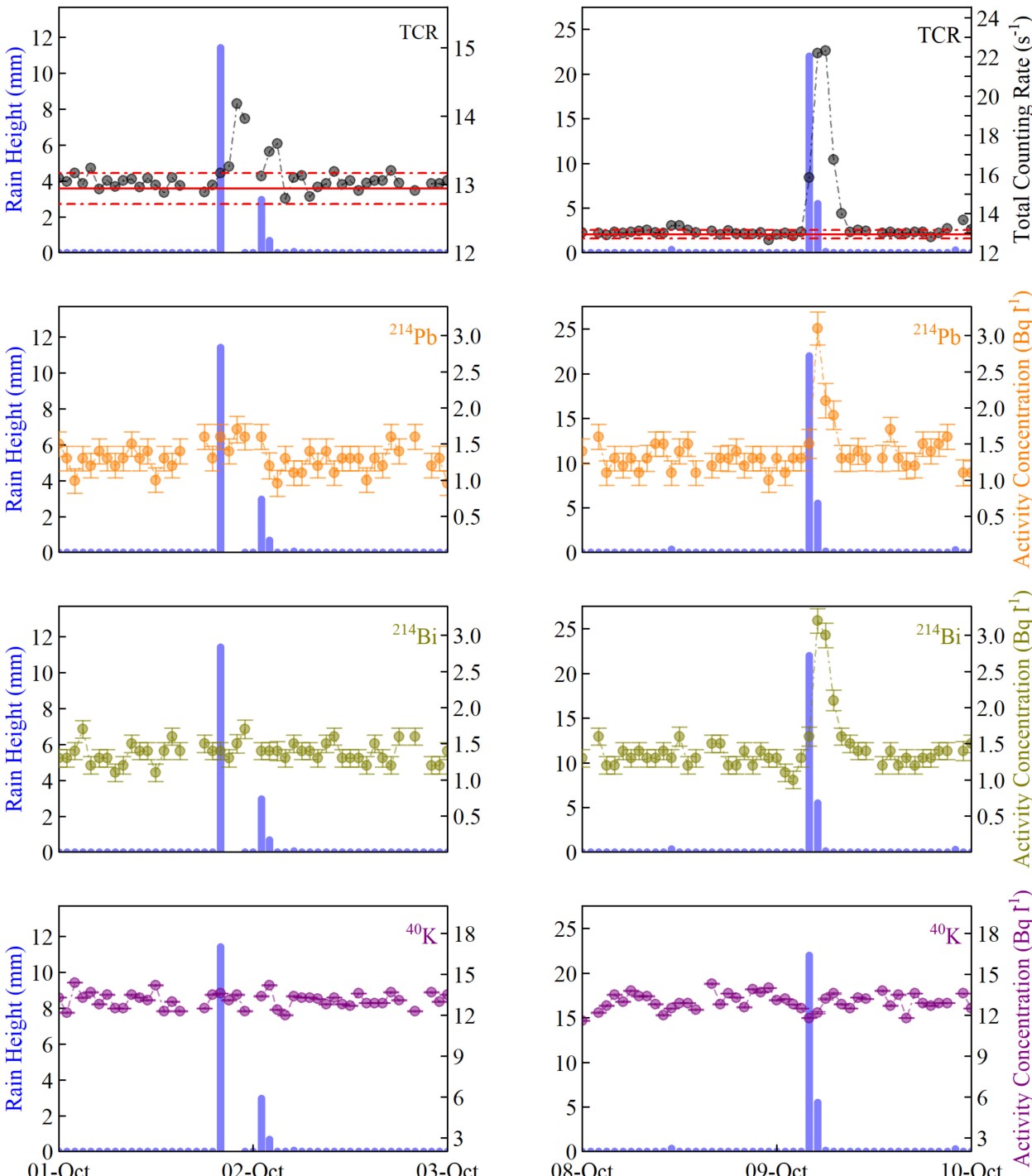

**Figure A5.** Representative rainfall events of October 2016. The *y*-axis of the first column regards rain height values (blue bars) while of the second TCR values, $^{214}$Pb, $^{214}$Bi and $^{40}$K activity concentration (circular points). In the subplots of TCR, the mean value during dry meteorological periods is depicted as a red solid line, and the range of the statistically significant criterion is depicted with red dotted lines.

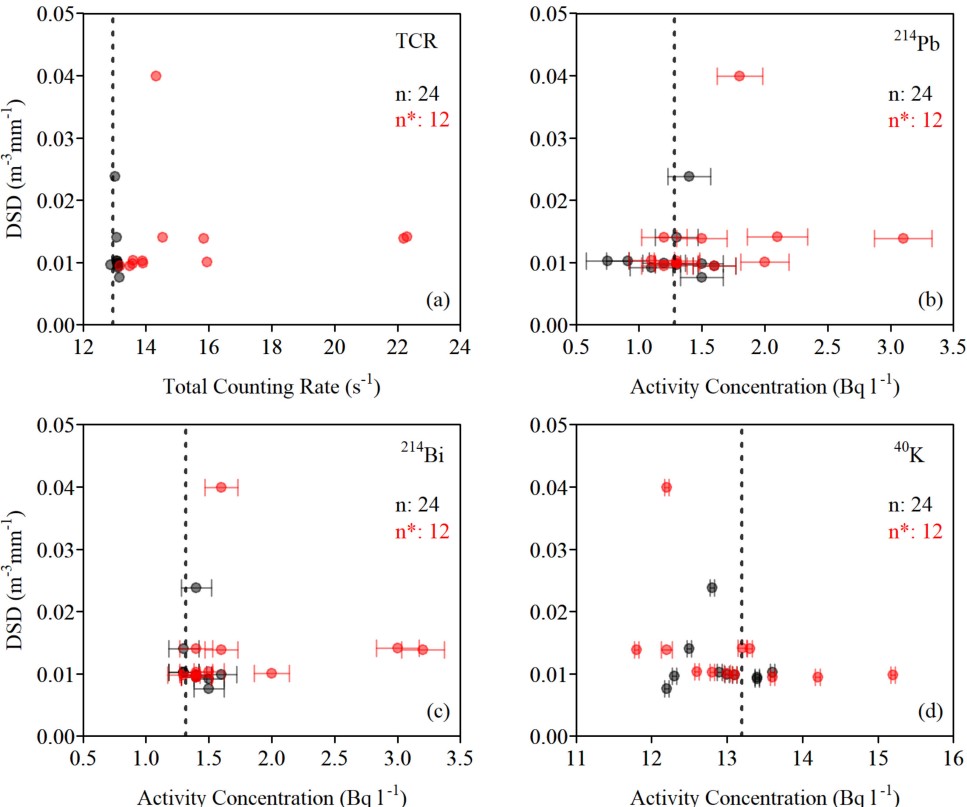

**Figure A6.** Drop size distribution of small size class (0.8–1.2 mm) over TCR (**a**), $^{214}$Pb (**b**), $^{214}$Bi (**c**) and $^{40}$K (**d**). The total number of events is *n* = 24. The events of statistically significant variation of TCR (*n*\* = 12) are presented with red dots.

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
