# Peer review of "Rainfall Investigation by Means of Marine In Situ Gamma-ray Spectrometry in Ligurian Sea, Mediterranean Sea, Italy"

_jmse, doi:10.3390/jmse9080903_

Round 1

Reviewer 1 Report

From my point of view, insufficient attention has been paid to the assessment of radionuclide fluxes with atmospheric fallout. In particular, the activity of beryllium-7 in atmospheric precipitation and seawater is quite high. Beryllium-7 can be easily measured using an isolated line. Perhaps this will be the next stage of the authors' research. The article is recommended for publication as presented.

Reviewer 2 Report

The article describes results of  measurements of natural radioactivity of marine water. The authors tried to link the rainfall events with variations of total activity of sea water measured by gamma-ray spectrometry. Additionally, they tried to find correlation of obtained data with air/cloud origin - terrestial/sea. The efforts paid by authors should be appreciated - huge numbers of measurements and data analysis were done. However, the main idea/theesis is not clear. Are you really want to detect rainfall events by use of gamma ray spectrometry - expensive and sophisticated instrumentation in comparision to conventional rainfall detectors? Probably the introduction should be improved and main idea should be clarified.

Questions/issues:

The measurement system is very briefly described. Please give more detail about detector: size, type, efficiency, etc.

What about calibration? How do you calcualte the radionuclide activity concentration. Is the self-attenuation significant in your case?

Why do you pay attention only to Bi-214/Pb-214 and K-40? Aren't Pb-212, Be-7 present as well in rain water? What about their contribution to total gamma radiation?

What about Pb-210? Despite the low gamma emission probability of Pb-210, its contribution may be significant due to high concentration... Did you check it? Is your detector sensitive to such low energy?

What informations can be read form charts in figure 10? Is there any relationship?

In my opinion, the most important weakness of this article is lack of reference to other radionuclides present in rainwater, like Pb-212 and Be-7. If you disagree, please comment why.

Round 2

Reviewer 2 Report

Thank you for your answers and corrections made in manuscript. I do not have any more questions/issues.